# Quantity-Sourced or Quality-Sourced? The Impact of Word-of-Mouth Recommendations on China Rural Residents’ Online Purchase Intention: The Chain Mediating Roles of Social Distance and Perceived Value

**DOI:** 10.3390/bs15121661

**Published:** 2025-12-02

**Authors:** Changxu Wang, Jinyong Guo

**Affiliations:** 1College of Economics and Management, Huaibei Normal University, Huaibei 235020, China; 13955473061@163.com; 2College of Economics and Management, Jiangxi Agricultural University, Nanchang 330045, China

**Keywords:** quantity-sourced, quality-sourced, word-of-mouth, social distance, perceived value, online purchase intention

## Abstract

This study examines how different types of word-of-mouth (WOM) influence online purchase intention (OPI) among rural residents, an area not yet fully explored. Based on social tie strength theory, we classify WOM into “quantity-sourced” (e.g., friends, family, general consumers) and “quality-sourced” (e.g., influencers, celebrities, professionals). We propose a chain mediation model involving social distance (SD) and perceived value (PV). Using survey data from 1005 rural residents in Jiangxi Province, China, and analyzing the data with structural equation modeling (SmartPLS 4), we find that quantity-sourced WOM positively affects OPI (β = 0.135), while quality-sourced WOM negatively affects it (β = −0.166). Mechanism analysis shows SD is a key mediator: quantity-sourced WOM shortens SD, thereby increasing OPI (β = 0.152), whereas quality-sourced WOM widens SD, reducing OPI (β = −0.047). PV mediates between quantity-sourced WOM and OPI (β = 0.043), but it shows no significant mediation between quality-sourced WOM and OPI (β = −0.002). Additionally, SD and PV serve as chain mediators between both types of WOM and OPI. These findings extend WOM theory to rural contexts and offer practical insights for governments and e-commerce platforms to develop differentiated WOM strategies and build localized WOM networks.

## 1. Introduction

As rural areas of China become increasingly informatized and e-commerce companies grow their presence in the rural market, online shopping is being incorporated into the quotidian lives of rural residents in China. In 2024, rural e-commerce sales (USD 403.14 billion) represented 18.5% of China’s total online retail volume (USD 2.18 trillion), which means that the online shopping consumption power of rural residents, who account for 33% of the country’s population, has not yet been fully released. Understanding the psychological drivers of their online purchase decisions is thus critical. This study focuses on how different types of word-of-mouth (WOM) recommendations influence rural residents’ online purchase intention (OPI) through the chain mediating roles of social distance (SD) and perceived value (PV).

Among the many factors affecting rural residents’ consumption psychology, WOM recommendations constitute pivotal drivers of rural consumers’ purchase decisions ([14]). It should be noted that rural areas of China, where collectivist culture prevails ([37]), whether online shopping or shopping in traditional stores, rural residents often consult other people’s evaluations before purchasing a product to decide whether to buy it ([53]). In particular, WOM recommendations from relatives, friends, and family members have high credibility ([28]). In the Internet era, rural residents with limited e-commerce literacy increasingly rely on diversified WOM channels when making online purchasing decisions, and the differences in the influence of different types of WOM sources on their online purchasing decisions are worth exploring; on the one hand, from WOM issued by opinion leaders (e.g., celebrities, Internet celebrities, product experts, etc.), and on the other hand, from WOM issued by ordinary people (e.g., customers, reviewers, friends, etc.) ([58]). However, are other people’s WOM recommendations always better?

In recent years, the exploration of WOM has been a research hot spot in the field of consumer psychology, and most of the existing literature has been developed from the perspectives of WOM information content ([43]), information environment ([56]), and information source ([18]). These studies predominantly examine characteristics of WOM information content and environment; for example, scholars have pointed out that the reliability of WOM sources and the number of online reviews in the information content characteristics impact consumers’ purchase intention ([12]; [52]). Additionally, scholars have compared effects across information environment characteristics, such as online WOM recommendations and offline WOM recommendations, on customers’ purchase intention ([3]). In terms of information source, some studies have explored how mass opinions and expert opinions influence the dissemination of health information ([29]). Additionally, some researchers have studied the differences in the impact of the quality-sourced and quantity-sourced health rumors in a health community scenario ([54]).

Regarding the participant profile, the existing literature on WOM has primarily examined urban contexts with weak-tie networks ([25]). In contrast, Chinese rural society is characterized by tight-knit strong ties, which may lead to fundamentally different psychological mechanisms in information dissemination ([8]). Specifically, in an environment dominated by strong ties, the relative effectiveness of quantity-sourced WOM (signaling majority choice) versus quality-sourced WOM (representing specialty) in reducing uncertainty and influencing purchase decisions may differ markedly from urban contexts.

Extant research on Word-of-Mouth (WOM) has provided robust findings. However, these findings are primarily based on studies in urban, weak-tie settings. Their applicability to strong-tie environments, such as rural China, remains unverified. This creates a key theoretical gap. First, it is unclear how different WOM sources influence consumers in strong-tie networks ([60]). Second, the psychological mechanism underlying this process is not well understood ([44]). Specifically, the chain mediating roles of Social Distance and Perceived Value between WOM and Online Purchase Intention have been largely overlooked. Examining this pathway is crucial to define the boundary conditions of existing theory and to explain the unique decision-making of rural consumers.

Therefore, this study attempts to integrate the above theories and aims to address the following questions by constructing a model of rural residents’ online purchase intention from the perspective of different WOM recommendation sources:What are the effects of different WOM recommendations on the rural residents’ OPI?Do SD and PV affect the rural residents’ OPI?Do SD and PV play a chain mediating role in WOM recommendation types and OPI?

To verify the influence difference between the two types of information sources, this study combines the theory of social tie strength from the perspective of WOM sources, and classifies WOM sources into “quantity-sourced” and “quality-sourced WOM” in accordance with the degree of strength of the social tie between information source and the individual. The research model of WOM Recommendations → SD → PV → rural residents’ OPI is constructed to explore how WOM recommendations influence the rural residents’ OPI through SD and PV.

The subsequent sections of this paper are structured as follows. Section 2 elaborates on the theoretical foundations and proposes specific hypotheses. Section 3 details the research design, measures, and data collection. Section 4 presents the findings of the empirical analysis. Section 5 presents the discussion and interprets the results. Section 6 elaborates on the theoretical and practical implications. Section 7 acknowledges the limitations and suggests directions for future research.

## 2. Theoretical Background and Research Hypothesis

### 2.1. Social Tie Strength Theory

This study employs Granovetter’s social tie strength theory as its theoretical framework. The theory conceptualizes interpersonal relationships as a continuum from strong to weak ties, distinguished across dimensions such as interaction frequency, emotional intensity, intimacy, and reciprocal services ([19]). These tie types play distinct roles in information diffusion: weak ties act as “bridges” to novel, heterogeneous information, while strong ties provide a “trust foundation” for influential messages. Building on this theoretical distinction, we operationalize online word-of-mouth into two types: quantity-based WOM and quality-based WOM.

For quantity-based WOM sources (e.g., family, friends, and numerous general consumer groups), it is easier to establish an emotional connection with rural consumers through strong ties ([22]), whose WOM communication is characterized by many sources of information but may be limited in specialization ([1]; [47]). For quality-based WOM sources (e.g., celebrities, Internet celebrities, and professional salespeople), these groups maintain weaker ties with rural consumers, while the content they disseminate is more specialized.

### 2.2. WOM Recommendations and Rural Residents’ Online Purchase Intention

With the development of social platforms, rural residents can access online shopping information from different information sharers through social media, and WOM recommendations can be categorized into quantity-sourced WOM and quality-sourced WOM based on tie strength between information sharers and consumers ([30]). For quality-sourced WOM, WOM publishers are groups of experts and opinion leaders in the field. Although some publishers make WOM recommendations to help others, they are considered to have a high probability of obtaining their benefits by publicizing the products for the enterprises; therefore, the strength of the tie between quality-sourced WOM and rural residents is low ([38]).

Quantity-sourced WOM refers to opinions or WOM that are agreed upon by the majority of people within a group—usually considered a common group ([30]; [48])—typically made up of customers (e.g., relatives, friends, most online reviewers) who have purchased ([17]), used, or experienced the product and who share their evaluations with other customers. The relationship between rural residents and publishers of quantity-sourced WOM was stronger than that of quality-sourced WOM ([6]).

Existing studies have shown that quantity-sourced WOM is essentially an endogenously generated self-organizing recommendation system, which forms a group consensus through multiple rounds of social validation by strong relational nodes ([42]); it has been verified and supervised by the surrounding relatives and friends or by the majority of ordinary consumers, and it is thus considered highly reliable and easier to spread and triggers rural consumers’ willingness to purchase online ([33]).

Quality-sourced WOM constitutes a signaling mechanism for institutional authority, and its information sources mainly include two types: (1) vertical experts with professional certifications and (2) institution-endorsed evaluators with contractual relationships with commercial entities ([49]).

Although some quality-sourced WOM recommenders make recommendations based on altruistic motives, frequent incidents of expert trust crises in digital marketing (celebrity endorsement scandals, influencer false marketing, etc.) have significantly increased institutional distrust among rural consumers ([4]).

According to the “2024 ‘November 11th Shopping Festival’ Consumer Rights Public Opinion Analysis Report” released by China Consumers’ Association, during the monitoring period (from 20 October to 16 November 2024), there were 129,000 negative comments related to “product quality,” reflecting that the quality of goods purchased online is a prominent issue. This lowered the rural residents’ trust in quality-sourced WOM recommenders. In addition, the e-commerce literacy of rural residents is relatively low ([46]); therefore, to reduce the cost of information processing, rural residents perceive quantity-sourced WOM as generating a higher level of altruistic motivation. Thus, quality-sourced WOM inhibits the online purchase intention of rural residents ([11]).

**H1.** 
*Quantity-sourced WOM positively affects rural residents’ OPI.*


**H2.** 
*Quality-sourced WOM negatively affects rural residents’ OPI.*


### 2.3. WOM Recommendations and Social Distance

Social distance (SD) is the core dimension of psychological distance, which refers to the degree of emotional and relational closeness between individuals. It reflects the degree to which individuals agree or are similar in specific attributes (e.g., emotional distance, status difference) during their interactions with others ([31]). This study defines social distance as a subjective sense of “separation in affection, intimacy, and social identity.” Operationally, it refers to the perceived psychological closeness between rural residents and the sender of word-of-mouth information ([55]).

Based on this, we hypothesize that quantity-sourced WOM is more likely to shorten social distance. This is because its sources, often ordinary people, share high social homogeneity with rural residents ([8]). Their shared experiences facilitate emotional connection and identity resonance. In contrast, a significant socioeconomic gap exists between quality-sourced WOM sources (e.g., influencers, celebrities) and rural residents. Corporate marketing can trigger resistance among rural consumers ([55]), reinforcing the “us versus them” group boundary and potentially increasing social distance.

Some scholars have confirmed that the quality and strength of the relationship between the WOM sender and the receiver affect social distance ([22]; [27]). This study hypothesizes that the quantity-sourced WOM is more likely to resonate with rural consumers through communication and interaction, thus bringing them closer to each other because the WOM senders are mostly the general public, belonging to the same group as rural residents. They share and evaluate the products they have used or know about from the customers’ perspective, thus bringing them closer to each other through communication and interaction ([55]; [59]). While the socioeconomic status gap between the quality-sourced WOM senders and rural residents is great, and enterprises may hire influencers and stars to gain profits and profit from selling to the fan base, rural residents have gradually become aware of these e-commerce platform routines and may have an aversion to the recommendations of the stars and influencers groups; this aversion can even create conflicts or contradictions that can then increase the social distance of rural residents ([16]).

**H3.** 
*Quantity-sourced WOM shortens SD for rural residents.*


**H4.** 
*Quality-sourced WOM increases SD for rural residents.*


### 2.4. WOM Recommendations and Perceived Value

Perceived value (PV), a core concept in marketing, refers to consumers’ internal evaluation of a product or service’s utility based on their own trade-offs between sacrifices (e.g., money, time) and benefits received (e.g., functional utility, emotional experience) ([57]). According to the Stimulus–Organism–Response theory, external stimuli (e.g., word-of-mouth recommendations) trigger cognitive and affective processes within consumers (organisms), subsequently influencing their behavioral responses ([15]). In shopping contexts, consumers leverage their prior knowledge to evaluate word-of-mouth information objectively, forming distinct perceptions of value.

Rural residents lack the necessary information and experience when online shopping and refer to WOM information from different groups ([8]). Because quantity-sourced WOM senders and rural consumers know each other well and have a certain foundation of trust, with a high degree of credibility and numerous and consistent opinions, rural residents obtain information on the quality, practicality, economy and other aspects of the product from groups similar to their own, thus increasing their understanding of the information on the products they buy ([8]). According to the consistency theory, consumers are more convinced of the product after being affirmed by their acquaintances, resulting in a higher perceived value ([35]).

For quality-sourced WOM, the current e-commerce market is saturated with excessive influencer-endorsed products, making it challenging for rural residents to evaluate source credibility due to information overload ([15]). When WOM sources are unverifiable or lack credibility, consumers’ perceived risk increases and their PV of the products diminishes.

**H5.** 
*Quantity-sourced WOM has a positive effect on rural residents’ PV.*


**H6.** 
*Quality-sourced WOM has a negative effect on rural residents’ PV.*


### 2.5. Social Distance and Purchase Intention

Social distance, as a key indicator of the degree of closeness between individuals or groups, is an essential antecedent variable for predicting consumer behavior ([2]). In the online purchase decision process, the social distance between consumers and sellers is increased because of the lack of face-to-face interaction. Therefore, social distance becomes an essential factor influencing online purchase intention ([32]). Some scholars believe that social distance can affect individual cognition and behavior, and when social distance is drawn closer, consumers’ sense of strangeness, disconnection, and alienation may be alleviated ([51]). Consumers’ confidence in the quality of goods are enhanced, which helps to improve consumers’ online purchase intention. In the context of online shopping, the absence of face-to-face communication renders SD a primary impediment. A diminished SD augments trust by fostering the perception of a relatable partner, consequently enhancing online purchase intention.

**H7.** 
*The closer the social distance, the stronger the rural residents’ online purchase intention.*


### 2.6. Perceived Value and Purchase Intention

Consumer PV is an essential determinant of purchase intention, and scholars have fully demonstrated the influence of PV on consumer purchase intention. Some studies show that PV affects consumers’ purchase intention, and the higher the PV, the stronger the consumers’ purchase intention ([36]; [45]). When rural residents highly recognize the products recommended by WOM recommenders when shopping online, rural residents’ willingness to purchase the products is also enhanced.

**H8.** 
*PV positively influences rural residents’ OPI.*


### 2.7. Social Distance and Perceived Value

As rural residents have long-developed offline shopping habits, when shopping on the Internet, the social distance between rural residents and product sellers is widened due to the distance of shopping, thus dissolving the psychological basis of cognition and trust in the traditional consumption model ([7]), increasing the consumer’s perceptual risk, and to a certain extent, reducing the consumer’s perceived value. On the contrary, a closer social distance produces a pleasant shopping experience and increases the PV of rural residents ([62]).

**H9.** 
*The closer the social distance, the higher the PV of rural residents when online shopping.*


### 2.8. The Chain Mediating Effect of SD and PV

In integrating the above insights, it can be noted that: (1) Different types of WOM can influence OPI not only indirectly by altering SD but also by changing PV. (2) More importantly, this study proposes a chain mediation model, where WOM influences OPI sequentially through SD and PV.

The theoretical rationale lies in a sequential “affective–cognitive” processing of information by consumers. Specifically, a change in SD first alters the emotional closeness and psychological affinity toward the WOM recommender (i.e., “willingness to trust” the source). This initial affective relationship and foundation of trust then sets the psychological context for subsequent value judgment. When rural residents perceive a closer SD with the WOM recommender, their feelings of unfamiliarity and uncertainty decrease significantly ([32]). This makes them more inclined to interpret product information positively, thereby enhancing their positive perception of the product’s usefulness and value for money (i.e., PV). Conversely, a wider SD amplifies distrust, which subsequently inhibits value perception ([55]). Therefore, SD operates first by shaping the psychological context for evaluation ([22]), while PV follows as the specific cognitive outcome formed within that context. Together, they constitute a continuous “affective–cognitive” mediation pathway.

**H10a.** 
*SD mediates the relationship between quantity-sourced WOM and rural residents’ OPI.*


**H10b.** 
*SD mediates the relationship between quality-sourced WOM and rural residents’ OPI.*


**H10c.** 
*PV mediates the relationship between quantity-sourced WOM and rural residents’ OPI.*


**H10d.** 
*PV mediates the relationship between quality-sourced WOM and rural residents’ OPI.*


**H10e.** 
*SD and PV play a chain mediating role between quantity-sourced WOM and rural residents’ OPI.*


**H10f.** 
*SD and PV mediate the chain between quality-sourced WOM and rural residents’ OPI.*


### 2.9. Theoretical Framework

This study examines the effects of WOM recommendations on rural residents’ OPI. Combined with the analysis of the related literature, WOM recommendations are divided into quantity-sourced WOM and quality-sourced WOM, and two mediating variables, SD and PV, are introduced to establish the research model, as shown in Figure 1.

## 3. Materials and Methods

### 3.1. Participants and Procedure

Study data come from the National Natural Science Foundation of China project “Research on the Identification of Differences in Online Shopping Expenditure Intention and Behavior of Rural Residents, Transformation Mechanisms and Spillover Effects” (No. 72063017). This project began in June 2021 and was completed in March 2022.

The target population was rural residents aged 18–75 in Jiangxi Province. ‘Rural residents’ were operationally defined as individuals with permanent household registration in villages or townships, residing in these areas for over 6 months per year.

The questionnaire consists of two parts: (a) respondents’ personal information, including gender, age, and education level, and (b) measurement items, including rural residents’ willingness to shop online through shopping websites/apps and psychological consumption traits. The survey area was chosen to be Jiangxi Province, a province in central China with a gross domestic product (GDP) of CNY 3207.47 billion in 2022, ranking 15th in China and at a medium level. Therefore, selecting rural residents in Jiangxi Province as the study sample represents all the provinces in central China well. The survey employed stratified random sampling; firstly, the per capita GDP of 100 counties in Jiangxi was ranked and stratified into three tiers, and, combined with the geographic location and rural population, a total of 20 sample counties and districts were selected in the three levels to carry out the field sampling research; secondly, 2–3 townships were selected in each sample county according to the per capita GDP; and then, 2–3 townships were identified in each township by economic development level and geographic location. Next, 2–3 administrative villages were identified in each township according to the level of economic development and geographic location. Finally, 8–10 rural residents were identified in each administrative town according to the equidistant step. At the same time, to ensure the validity of the questionnaire, the respondents were told in advance that if they answered the questionnaire thoughtfully, they would receive a small gift after answering. The survey was conducted through face-to-face, on-site interviews by trained investigators across the 11 prefecture-level cities of Jiangxi Province. To ensure data quality and mitigate common method bias, several procedural remedies were implemented: (1) Respondents were assured of anonymity and confidentiality; (2) measurement items for different constructs were separated in the questionnaire; and (3) some items were reverse coded to prompt careful reading. The research issued 1080 questionnaires and received 1028 questionnaires, eliminating any data-defective samples to get 1005 valid questionnaires, with a questionnaire validity of 97.76% and good representation.

### 3.2. Variable Measurement

To ensure measurement validity in the rural Chinese context, a rigorous translation and back-translation procedure was implemented. First, two bilingual researchers independently translated the original English scales into Chinese. Then, a third bilingual expert, blinded to the original version, performed back-translation. The research team compared the back-translated version with the original, discussing and resolving any semantic discrepancies until consensus was achieved. This procedure ensured conceptual equivalence and cultural appropriateness of the scales, establishing a reliable foundation for subsequent empirical analysis.

This study employed a cross-sectional survey design to investigate the proposed research model. We used exploratory factor analysis (EFA) during the pilot study for scale refinement. A pilot study with 100 valid responses preceded the formal survey. We performed exploratory factor analysis (EFA) on the questionnaire items using SPSS26 to assess validity and reliability. After deleting the items with cross-loading and factor loading less than 0.500, the scale finally retained 15 items and 5 factors, with a total variance of 76.35%. As shown in Table 1, this study’s questionnaire design draws on previous research and combines the rural residents’ characteristics in the online purchasing process to design the scale, using a 5-point Likert scale (1 = strongly disagree to 5 = strongly agree).

## 4. Results

### 4.1. Respondent Demographic Characteristics

This paper used SPSS26 software to analyze the descriptive statistics of 1005 valid questionnaires, as indicated in Table 2. Among the 1005 rural residents interviewed, men and women accounted for half of the respondents, and the age distribution of the rural residents aged 46–60 and over 65 years old was the greatest, accounting for 34.7% and 22.2%, respectively, which shows that most of the interviewees were middle-aged and older adults. The characteristics of such an age distribution are consistent with the aging trend in China. Such age distribution characteristics coincide with the aging trend occurring in rural areas. The self-assessed health status of rural residents is good; the majority of respondents, 62.4%, have an educational level below junior high school, which is also consistent with the overall low educational level of rural residents surveyed by the China Statistics Bureau. Finally, this group’s average annual disposable income is mainly concentrated in the range of CNY 10,000–50,000, accounting for 54.43%, which shows that rural residents still have a certain level of purchasing power. Therefore, the sample distribution of this study is reasonable enough to provide a guarantee for further study.

### 4.2. Reliability and Validity Analysis

From the measured results of the model in Table 3, the standardized factor loading for each item is greater than the threshold value of 0.5. Usually, the value of Cronbach’s α coefficient is between 0 and 1. If the coefficient does not exceed 0.6, it is generally considered that the internal consistency reliability is insufficient, and in practical research, it is usually considered that greater than 0.6 is sufficient. Cronbach’s α values in this study are all greater than the threshold value of 0.6 and, therefore, meet the requirements. Secondly, the reliability of the questionnaire is examined using the composite reliability (CR) value, and the CR values are all in the range of 0.808–0.947; this shows that the indicators of the dimensions of the variables have sufficient reliability and internal consistency. The average variance extracted (AVE) is an essential indicator of convergent validity; the AVE values of the scale were all greater than the reference value of 0.5. Therefore, the scale has high validity.

### 4.3. Discriminant Validity Analysis

PLS-SEM can be used to determine the discriminant validity of the model in two ways. The first method is the Fornell–Larcker criterion ([23]), which judges that the correlation coefficients between latent variables should be less than the root value of their respective AVE; as shown Table 4, the correlation coefficients between all the latent variables satisfy the above requirements. The second method is by whether the value of HTMT (Heterotrait–Monotrait Ratio) is less than 0.85 ([21]); as shown Table 5, the HTMT values among the latent variables are all less than 0.85. Therefore, the discriminant validity of the model passed the test.

### 4.4. Co-Linear Analysis

Co-linear analysis is an essential step in judging the structural model ([26]). The diagnosis criterion is that the VIF value of the internal model is less than 5. As shown in Table 6, the VIF value is between 1.002 and 2.503—less than the critical value of 5. Therefore, it can be judged that there is no multi-covariance problem between the variables, and the model passes the covariance diagnosis.

### 4.5. Path Analysis

The structural model’s path coefficients were estimated using partial least squares structural equation modeling in SmartPLS4. A bootstrap procedure with 5000 subsamples was used for significance testing. The criterion for significant judgment of the path coefficients is that when the t-value of the path coefficients is greater than 1.96, it means that the path coefficients pass the test at the 5% level. The model demonstrated substantial explanatory power, accounting for 45.7% of the variance in online purchase intention (R^2^ = 0.457), 33.2% in social distance (R^2^ = 0.332), and 41.6% in perceived value (R^2^ = 0.416). The standardized root mean square residual (SRMR) was 0.045, indicating a good model fit. The results of the primary effects test and mediation effects test in this study are shown in Table 7 and Figure 2: (1) Quantity-Sourced (β = 0.135, t = 4.184, *p* = 0.000), SD (β = 0.349, t = 8.267, *p* = 0.000), and PV (β = 0.130, t = 3.161, *p* = 0.002) have a significant positive effect on OPI. All have a significant positive effect, and Quality-Sourced (β = −0.166, t = 6.667, *p* = 0.000) has significant negative effect on OPI; therefore, hypotheses H1, H2, H7, and H8 are verified. The corresponding effect sizes (f^2^) for these paths on OPI are 0.035 (small) for Quantity-Sourced WOM, 0.058 (small) for Quality-Sourced WOM, 0.150 (medium) for SD, and 0.025 (small) for PV. (2) Quantity-Sourced (β = 0.435, t = 15.678, *p* = 0.000) has a significant positive effect on SD, and Quality-Sourced (β = −0.083, t = 2.925, *p* = 0.003) has a significant negative effect on SD; therefore, hypotheses H3 and H4 are validated. (3) Quantity-Sourced (β = 0.328, t = 12.539, *p* = 0.000) and SD (β = 0.571, t = 27.722, *p* = 0.000) have a significant positive effect on PV, while Quality-Sourced (β = −0.018, t = 0.885, *p* = 0.376) has no significant effect on PV; therefore, hypotheses H5 and H9 are tested, and hypothesis H6 is not tested.

In the mediation effects test: (1) SD mediates between Quantity-Sourced and OPI (β = 0.152, t = 7.611, *p* = 0.000) and between Quality-Sourced and OPI (β = −0.047, t = 2.877, *p* = 0.004), with the hypotheses H10a and H10b holding. (2) PV mediates between Quantity-Sourced and OPI (β = 0.043, t = 3.052, *p* = 0.002), whereas PV does not mediate between Quality-Sourced and OPI (β = −0.002, t = 0.803, *p* = 0.422), so hypothesis H10c is valid, and H10d is not validated. (3) There is a chain mediation effect between SD and PV between Quantity-Sourced and OPI (β = 0.043, t = 3.052, *p* = 0.002) and between Quality-Sourced and OPI (β = −0.006, t = 2.098, *p* = 0.036); therefore, hypotheses H10e and H10f are valid.

## 5. Discussion

Prior research has centered on the content and recipient characteristics in WOM recommendations while paying insufficient attention to the information sources of WOM recommendations. In particular, few investigations have assessed how WOM recommendations influence rural residents’ OPI, nor have they comprehensively considered how external WOM characteristics act on consumers’ OPI through internal psychological perceptions ([32]). Grounded in Granovetter’s social tie strength theory, this study divides WOM recommendations into quantity-sourced WOM and quality-sourced WOM ([8]). The findings show their different effect paths. This deepens our understanding of how strong and weak ties work in rural digital consumption decisions. First, quantity-sourced WOM most strongly improves OPI. This supports Granovetter’s view that strong ties are a “trust foundation.” In the close rural setting, information from trusted sources remains key to reducing uncertainty ([19]). Second, quantity-sourced WOM effectively shortens the Social Distance (SD) between rural residents and the online world. This increases their OPI. However, quality-sourced WOM significantly increased SD. This suggests that building trust through weak ties requires first closing the social gap with rural residents ([59]). In summary, strong- and weak-tie theory remains powerful in explaining rural digital behavior. However, it now works through new signals like quantity-sourced and quality-sourced WOM. These reshape rural residents’ decisions through internal paths like SD and PV.

First, quantity-sourced WOM, SD, and PV positively affect rural residents’ OPI. In contrast, quality-sourced WOM suppresses rural residents’ OPI, similar to the study by Jucks et al. on the decision of majority opinion and single expert opinion on people’s adoption of health information ([30]). In addition, the analysis results further demonstrate that SD has the most significant effect on rural residents’ OPI. Thus, SD between rural residents and e-commerce sellers significantly affects their purchase intention under conditions of online distance shopping. Additionally, a body of research has verified that PV facilitates OPI ([39]), and the same result is obtained in this paper.

Second, the existing literature has not yet reached consistent conclusions on the relationship between different WOM recommendation types and social distance, and there is a particular lack of empirical testing for rural arenas ([5]); this study shows that quantity-sourced WOM contributes positively to SD, while quality-sourced WOM has a negative effect on SD. This result suggests that recommendation behaviors based on strong social ties tend to be trusted in rural China, and that the homogeneous social attributes (e.g., similar living backgrounds, consumption habits) and emotional trust bases between quantity-sourced WOM publishers and rural residents make their recommendations more likely to be perceived as altruistic sharing; on the contrary, the differences in recommendation behaviors between authoritative sources, such as celebrities, have significant socioeconomic status gaps with rural residents ([9]). Their recommendation behaviors are often attributed to the drive of commercial interests, and this kind of self-interested attribution weakens the perceived reliability of the sources, thus expanding the SD. Quantity-sourced WOM positively affects PV, and quality-sourced WOM exerts no effect on PV; however, the path coefficient is still negative. As an old Chinese saying goes, “A gold cup is not as good as the people’s word of mouth”, and rural residents prefer the WOM of ordinary people. Descriptive analysis of the research sample reveals that more than half of the rural areas are middle-aged and old-aged groups, and when making online purchases, most rural residents perceive more unknown risks because they are not familiar with online purchases ([24]). Therefore, they prefer recommendations from people familiar and similar to them to enhance their perceived value rather than from celebrities.

The results additionally revealed that SD contributes positively to PV. The closer the SD between rural residents and the WOM recommender, the higher the PV, which is consistent with the finding of Kim et al. that “SD negatively predicts product evaluation ([34])”. However, this study further extends it to rural consumption scenarios and verifies the direct effect of SD on the PV of online purchases by rural residents.

Finally, mediation analysis revealed that SD plays a mediating role in the influence of quantity-sourced WOM and quality-sourced WOM on rural residents’ OPI, respectively. Quantity-sourced WOM publishers increase rural residents’ OPI by bringing them closer to the SD, while quality-sourced WOM increases the SD between them and rural residents and decreases their OPI ([37]). This validates the importance of SD in the rural residents’ OPI. In addition, there is a chain mediation effect of “WOM recommendation → SD → PV → rural residents’ OPI” between the type of WOM recommendation (quantity-sourced WOM vs. quality-sourced WOM) and rural residents’ OPI, which reveals the dual paths of SD as a “trust transfer bridge” and PV as a “behavioral decision pivot”. The specific path of action is as follows: quantity-sourced WOM publishers increase the value perception of rural residents and promote the formation of purchase intention by narrowing the SD with rural consumers. However, the opposite relationship exists for quality-sourced WOM ([50]). This finding validates the mediating mechanism between SD and PV in rural online consumption scenarios and expands the boundaries of the application of SD in the sinking market.

The research model reveals the transmission mechanism of WOM Recommendations → Social Distance → Perceived Value → rural residents’ Online Purchase Intention, which is of great significance for e-commerce platforms to formulate differentiated WOM communication strategies based on the differences in social attributes of rural residents and to construct “localized word-of-mouth networks” by enhancing the effect of SD through geographic and kinship relationship chains.

## 6. Implications

### 6.1. Theoretical Implications

Rural consumers are often exposed to two types of WOM information when shopping for products, namely, quantity-sourced WOM (e.g., friends, family, and many general consumer groups) and quality-sourced WOM (influencers, celebrities, and professional salespeople), and drawing on Granovetter’s social tie strength theory, we develop a structural equation model to examine how bidirectional WOM recommendations influence rural consumers’ OPI. This study has the following theoretical significance: (1) the introduction of “social tie strength” into the analytical framework of rural residents’ online purchasing intention, which provides a theoretical anchor point for the understanding of “relational consumption” in rural areas by examining the social tie strengths of the IWOM publishers and recipients. It expands the application boundaries of the theory of social tie strength, constructs a framework for analyzing quantity-sourced and quality-sourced WOM based on the dual perspectives of strong and weak relationships, and systematically reveals the mechanism of the influence of WOM source types on rural residents’ OPI under different social relationship strengths. (2) Through structural equation modeling path analysis, the study reveals the causal chain of “WOM Recommendations → Social Distance → Perceived Value → rural residents’ OPI”, which proves that social distance is not only a carrier of trust transmission but also indirectly affects purchase decisions by influencing value judgment. (3) By overcoming the limitations of existing studies focusing on urban residents and young groups, we analyze the cognitive differences in WOM information processing of rural residents with relatively limited Internet literacy and a weak foundation for trust construction and improve the theoretical mapping of the influence effect of WOM recommendations in the context of the digital divide.

### 6.2. Practical Implications

Furthermore, these findings offer practical implications for both policymakers and e-commerce marketers.

First, given the negative impact of quality-sourced WOM (e.g., from experts or celebrities) on rural residents’ online purchase intention, online sellers should reconsider influencer selection for this market. They should prioritize relatable local influencers over distant experts to reduce the perceived social distance. This approach can mitigate the negative effect identified in our study.

Second, to leverage the positive influence of quantity-sourced WOM (e.g., from family and friends), platforms should develop features that encourage sharing and displaying this type of WOM. For instance, optimize group buying features to make it easier for users to share shopping links directly within private chat groups (e.g., on WeChat) for family and friends to collaboratively purchase.

Finally, given rural residents’ reliance on familiar social networks, establishing village e-commerce service stations that facilitate shopping through trusted community members can effectively bridge the online trust gap.

## 7. Limitations and Future Research

Owing to external constraints, this study still has certain shortcomings. First, the thesis explores the impact of quantity-sourced WOM and quality-sourced WOM on rural residents’ online purchase intention. However, in addition to these two types of WOM information, additional WOM information influences rural residents’ OPI; for example, alongside social media’s advancement, internet WOM has become a hot spot in the field of marketing. Future research could explore how these media-based word-of-mouth forms influence rural residents’ purchase decisions by examining their roles in fostering a sense of presence and authenticity ([10]; [13]; [41]; [50]).

Regarding the completeness of the theoretical model, this study reveals the chain mediating roles of SD and PV. However, it does not fully consider potential boundary conditions that may moderate this process. For instance, according to [61] ([61]), product type (utilitarian vs. hedonic) is likely to moderate the strength of the WOM effect ([61]). Future research could systematically examine key moderating variables such as product type, consumer innovativeness, and cultural–regional differences. This would help clarify the scope and theoretical boundaries of our findings.

The reliance on self-reported data poses a limitation, as it is potentially influenced by social desirability and common method variance. Future investigations should integrate self-reports with objective behavioral data (e.g., click-through rates, actual purchases) from online purchases. This approach would significantly improve the validity and robustness of the research.

## Figures and Tables

**Figure 1 behavsci-15-01661-f001:**
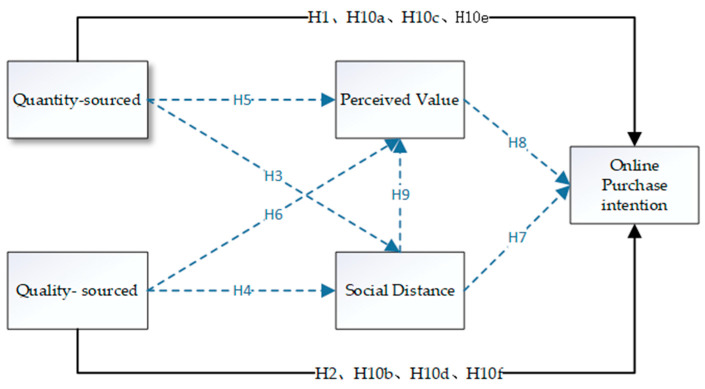
Model of the impact of WOM recommendations on rural residents’ online purchase intention.

**Figure 2 behavsci-15-01661-f002:**
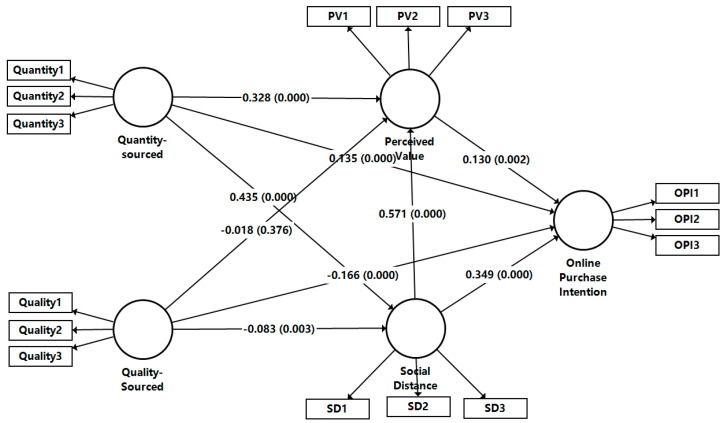
Analytical result of the model.

**Table 1 behavsci-15-01661-t001:** Questionnaire items and adoption sources.

Variables	Measurement Items	Adopted From
Quantity-Sourced	Throughout electronic shopping decision processes, you are significantly more inclined to refer to your family’s recommendation. Throughout electronic shopping decision processes, you are significantly more inclined to refer to your relatives’, friends’, or neighbors’ recommendations. Throughout electronic shopping decision processes, you are significantly more inclined to the general public’s recommendation.	([8]; [30])
Quality-Sourced	When making online shopping decisions, you show preference for the recommendations of salespersons. When making online shopping decisions, you show preference for the recommendations of online celebrities or stars. When making online shopping decisions, you show preference for the recommendations of industry experts.	([8]; [20])
Social Distance	I feel that the lifestyle of online shopping recommenders is similar to mine. I belong to the same social circle as those who frequently post online shopping recommendations. I am willing to actively establish social connections with those whom I trust in online shopping recommendations.	([62])
Perceived Value	You can buy more cost-effective goods through online shopping. You can buy better quality goods through online shopping. You can purchase satisfactory goods or services through online shopping.	([58])
Online Purchase Intention	In the future, when you need something, you will tend to do online shopping. In the future, you will make more frequent online purchases. In the future, the amount of money you spend on online shopping will be greater.	([40])

**Table 2 behavsci-15-01661-t002:** Demographic characteristics (*n* = 1005).

Variables		Frequency	Percentage
Gender	Male	541	53.8
	Female	464	46.2
Age	18–30	190	18.9
	31–45	243	24.2
	46–60	349	34.7
	Over 65	223	22.2
Health Condition	Very unhealthy	15	1.5
	Not very healthy	32	3.2
	Good	143	14.3
	Relatively healthy	382	38
	Very healthy	433	43.1
Education	Primary school and below	282	28.1
	Junior high school	344	34.3
	Technical secondary school or high school	307	30.5
	Junior college	206	20.5
	Undergraduate	68	6.8
	Postgraduate	80	8.0
Annual Income	CNY 0–10,000	303	30.15
	CNY 10,001–50,000	547	54.43
	CNY 50,001–100,000	120	11.94
	Over CNY 100,000	35	3.48

**Table 3 behavsci-15-01661-t003:** Reliability and validity analysis.

Constructs	Item	Factor Loading	Cronbach’s α	CR	AVE
Quality-Sourced	Quality1	0.528	0.682	0.81	0.598
	Quality2	0.896			
	Quality3	0.845			
Quantity-Sourced	Quantity1	0.834	0.654	0.808	0.585
	Quantity2	0.745			
	Quantity3	0.709			
Social Distance	SD1	0.921	0.886	0.929	0.815
	SD2	0.912			
	SD3	0.874			
Perceived Value	PV1	0.875	0.825	0.896	0.741
	PV2	0.883			
	PV3	0.824			
Online Purchase Intention	OPI1	0.949	0.916	0.947	0.856
	OPI2	0.951			
	OPI3	0.874			

**Table 4 behavsci-15-01661-t004:** Discriminant validity (FORNELL).

	OPI	PV	Quality-Sourced	Quantity-Sourced	SD
OPI	0.925				
PV	0.473	0.861			
Quality-Sourced	0.37	0.579	0.774		
Quantity-Sourced	−0.22	−0.091	−0.045	0.765	
SD	0.518	0.716	0.438	−0.102	0.903

**Table 5 behavsci-15-01661-t005:** Discriminant validity (HTMT).

	OPI	PV	Quality-Sourced	Quantity-Sourced	SD
OPI					
PV	0.539				
Quality-Sourced	0.419	0.713			
Quantity-Sourced	0.274	0.127	0.156		
SD	0.571	0.835	0.468	0.121	

**Table 6 behavsci-15-01661-t006:** VIF value of the inner model matrix.

	OPI	PV	Quality-Sourced	Quantity-Sourced	SD
OPI					
PV	2.503				
Quality-Sourced	1.507	1.238			1.002
Quantity-Sourced	1.011	1.011			1.002
SD	2.064	1.249			

**Table 7 behavsci-15-01661-t007:** Path analysis.

	β	Standard Deviation	*t*	*p*	LLCI	ULCI	Decision
H1: *Quantity-Sourced → OPI*	0.135	0.032	4.184	0.000	0.072	0.200	Supported
H2: *Quality-Sourced → OPI*	−0.166	0.025	6.667	0.000	−0.214	−0.119	Supported
H3: *Quantity-Sourced → SD*	0.435	0.028	15.678	0.000	0.380	0.489	Supported
H4: *Quality-Sourced → SD*	−0.083	0.028	2.925	0.003	−0.141	−0.029	Supported
H5: *Quantity-Sourced → PV*	0.328	0.026	12.539	0.000	0.276	0.378	Supported
H6: *Quality-Sourced → PV*	−0.018	0.020	0.885	0.376	−0.057	0.021	Unsupported
H7: *SD → OPI*	0.349	0.042	8.267	0.000	0.263	0.429	Supported
H8: *PV → OPI*	0.130	0.041	3.161	0.002	0.048	0.209	Supported
H9: *SD → PV*	0.571	0.021	27.722	0.000	0.531	0.611	Supported
H10a: *Quantity-Sourced → SD → OPI*	0.152	0.020	7.611	0.000	0.114	0.193	Supported
H10b: *Quality-Sourced → SD → OPI*	−0.047	0.016	2.877	0.004	−0.082	−0.017	Supported
H10c: *Quantity-Sourced → PV → OPI*	0.043	0.014	3.052	0.002	0.015	0.070	Supported
H10d: *Quality-Sourced → PV → OPI*	−0.002	0.003	0.803	0.422	−0.009	0.003	Unsupported
H10e: *Quantity-Sourced → SD → PV → OPI*	0.043	0.014	3.052	0.002	0.012	0.053	Supported
H10f: *Quality-Sourced → SD → PV → OPI*	−0.006	0.003	2.098	0.036	−0.013	−0.001	Supported

## Data Availability

The data presented in this study are available upon request from the corresponding author due to privacy restrictions.

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
