# Peer review of "Quantity-Sourced or Quality-Sourced? The Impact of Word-of-Mouth Recommendations on China Rural Residents’ Online Purchase Intention: The Chain Mediating Roles of Social Distance and Perceived Value"

_behavsci, 2025, doi:10.3390/bs15121661_

Round 1
Reviewer 1 Report
Comments and Suggestions for Authors
Dear authors, I hope my comments will be useful and will let you improve the article.
- The authors use the terms quality and quantity of Word-of-Mouth (WOM); however, these terms are not yet commonly standardized in the existing literature. Although the authors explain these terms in the literature review section, the cited sources (Jucks and Thon 2017; Shi and Wojnicki 2014), do not use these exact terms, which raises concerns about consistency and clarity. I suggest they either use more common terms or clearly justify and define these concepts with supporting references to improve clarity.
- Since the readers of this article are international scholars, it would be clearer to present the amount in USD rather than just Chinese Yuan. Alternatively, providing both the original currency and its equivalent in USD would be most helpful for a global audience.
- The introduction section does not clearly explained the research gaps or emphasize the significance of the study. For example, the authors mention “The existing studies have not examined the effects of quantity-sourced and quality-sourced WOM on rural residents’ online purchase intentions.” on line 85of page 2. However, this alone - focusing only on rural populations - cannot be considered a valid research gap.
- In the Theoretical Background section, the authors should include a detailed paragraph explaining how Granovetter’s concept of social tie strength serves as the baseline theory for this study. Additionally, in the Discussion section, they need to interpret their findings in relation to this theory to strengthen the connection between theory and results.
- The authors, in their definition of social distance (page 4, line 173), cite Akerlof (1997). While Akerlof’s paper, “Social Distance and Social Decisions” (1997), presents a model of social distance from an economic perspective—focusing on how agents interact more closely when socially “closer” and less when more distant—I could not find any passage in the paper that matches the authors’ wording regarding emotional closeness, attribute similarity, and related concepts.
- The authors need to provide stronger and more detailed justifications for their supporting hypotheses in the Theoretical Background section. Additionally, many of the references they use are not directly related to WOM studies, which weakens the foundation of their arguments. In addition, for hypothesis H3, the authors stated that “Quantity-sourced WOM decreases social distance (SD) for rural residents.” However, the results showed a positive effect, which the authors accepted despite contradicting the original hypothesis.
- In some paragraphs (p. 4, lines 180–190; p. 5, lines 203–211), the authors rely on a single citation. If these paragraphs include broader claims beyond what the cited source covers, additional references should be provided to adequately support those arguments.
- The authors introduce the chain mediating effect of SD and PV without clearly explaining its rationale or theoretical basis.
- In the methodology section, under Variable Measurement, the authors do not specify whether they used the original scales in English or in Chinese. If the scales were translated into Chinese, they should explain the translation process used to minimize bias and maintain the validity of the measures.
- In Table 1, the authors cite Jucks and Thon (2017) for the Quantity-sourced WOM scale, but their study and items do not align with this concept. Similarly, Hendriks et al. (2015) is cited for the Quality-sourced scale, yet their items do not match those used current study. This raises concerns about the relevance of the sources and the validity of the measures.
- The authors should provide citations for all thresholds used in the analysis and report the R² values for the regression paths.
- The discussion section needs to be improved by providing clear justifications for the results related to each hypothesis and by more effectively supporting the study’s findings with evidence and references from previous research.
- Practical implications should logically follow from the data and results presented in the study. For example, 'prohibit influencers and stars from recommending unused goods, educated sales staff to provide timely and professional answers' are not studied in this study.
Some sentences are too long and hard to follow, and punctuation is sometimes missing or incorrect. Fixing these issues will make the writing much clearer and easier to understand.
Author Response
Comment1:
The authors use the terms quality and quantity of Word-of-Mouth (WOM); however, these terms are not yet commonly standardized in the existing literature. Although the authors explain these terms in the literature review section, the cited sources (Jucks and Thon 2017; Shi and Wojnicki 2014), do not use these exact terms, which raises concerns about consistency and clarity. I suggest they either use more common terms or clearly justify and define these concepts with supporting references to improve clarity.
Response:
This study investigates the impact of word-of-mouth (WOM) by focusing on its two fundamental dimensions: WOM quantity and WOM quality. We referred to the papers published by Chen et al(2023) in important journal.They explicitly categorizes and defines 'quantity-sourced WOM' and 'quality-source WOM' . Accordingly, in this study:
WOM quantity refers to the volume or amount of WOM messages a potential consumer is exposed to.
'WOM quality' refers to the perceived credibility, usefulness, and informational value contained in the WOM messages.
References
Chen C, Zhang D, Zhang F and Gao H (2023) Is professional reputation better than public reputation? A study on the influence of different types of reputation on customers' purchase intentions. Nankai Business Review 26(4).
Comment2:
Since the readers of this article are international scholars, it would be clearer to present the amount in USD rather than just Chinese Yuan. Alternatively, providing both the original currency and its equivalent in USD would be most helpful for a global audience.
Response:
We agree that providing an equivalent amount in US dollars will significantly enhance the clarity of the financial data and the readability for international readers. Therefore, we have revised the manuscript. In the first instance of the main text, both the original RMB amount and the equivalent US dollar amount are presented together. These conversions are based on the average annual exchange rate published by the People's Bank of China.
Comment3:·
The introduction section does not clearly explained the research gaps or emphasize the significance of the study. For example, the authors mention “The existing studies have not examined the effects of quantity-sourced and quality-sourced WOM on rural residents’ online purchase intentions.” on line 85of page 2. However, this alone - focusing only on rural populations - cannot be considered a valid research gap.
Response:
We sincerely thank the reviewer for this critical insight. We agree that merely stating the rural context is insufficient. Following your suggestion, we have thoroughly revised the introduction to reframe the research gap. We now emphasize that the gap stems from the theoretical interplay between types of WOM and the distinct social structure (strong-tie networks, high homogeneity) of rural communities, which may challenge or extend the existing theories built upon urban contexts. The revised manuscript clarifies that our goal is not just to study a new population, but to test the boundary conditions of existing theories and uncover the unique psychological mechanisms (e.g., through social distance) at play in this understudied yet important setting.
Comment4
- In the Theoretical Background section, the authors should include a detailed paragraph explaining how Granovetter’s concept of social tie strength serves as the baseline theory for this study. Additionally, in the Discussion section, they need to interpret their findings in relation to this theory to strengthen the connection between theory and results.
Response:
As suggested, we have made significant revisions to the theoretical background section and the discussion section.
In the theoretical background section, we have added a detailed discussion that clearly links Granovetter's theory of strong/weak ties with our empirical operation of "quantity source" and "quality source" word-of-mouth information, explaining their respective roles as "bridges" in weak ties and as "trust bases" in strong ties.
In the discussion section, we now systematically interpret our key findings through this foundational theory. We explain how the effectiveness of high-quality source word-of-mouth information aligns with the power of strong ties, and how the mechanism of quantity source word-of-mouth information reflects the connection function of weak ties, thereby constructing a coherent narrative that closely links our research results to the theoretical framework.
Comment5
The authors, in their definition of social distance (page 4, line 173), cite Akerlof (1997). While Akerlof’s paper, ”Social Distance and Social Decisions” (1997), presents a model of social distance from an economic perspective—focusing on how agents interact more closely when socially “closer” and less when more distant—I could not find any passage in the paper that matches the authors’ wording regarding emotional closeness, attribute similarity, and related concepts.
Response:
We sincerely thank the reviewer for this exceptionally insightful and correct observation. We acknowledge that our initial use of Akerlof (1997) was not optimally aligned with the socio-psychological dimensions of social distance that we actually operationalized in our hypotheses. Following the reviewer's guidance, we have revised the manuscript to root the definition of social distance firmly within the socio-psychological literature.
Specifically, in the hypothesis development section, we have now:
Explicitly defined social distance as a subjective perception of "separation in affection, intimacy, and social identity" based on classical works (e.g., Park, 1924; Bogardus, 1933).
Clarified the theoretical focus of our study on this subjective perceptual dimension, distinguishing it from Akerlof's economic model.
Ensured complete consistency between this refined definition and the subsequent logic in our hypothesis development, which revolves around concepts of emotional connection.
Comment6
The authors need to provide stronger and more detailed justifications for their supporting hypotheses in the Theoretical Background section. Additionally, many of the references they use are not directly related to WOM studies, which weakens the foundation of their arguments. In addition, for hypothesis H3, the authors stated that “Quantity-sourced WOM decreases social distance (SD) for rural residents.” However, the results showed a positive effect, which the authors accepted despite contradicting the original hypothesis.
Response:
We sincerely thank the reviewer for raising this point. We apologize for the ambiguity in the description of the variable direction for "social distance" in our original manuscript. In this study, "social distance" was measured on a 1-5 Likert scale, where a higher score indicates a closer (or lesser) social distance. Therefore, to ensure complete consistency between our textual descriptions and the measurement scale, we have standardized all relevant descriptions in the revised manuscript: we now uniformly use "shortens social distance" to describe an increase in scores and "lengthens social distance" to describe a decrease. Specifically, Hypothesis H3 has been revised to state: "Quantity-source word-of-mouth shortens the social distance of rural residents."
Comment7
- In some paragraphs (p. 4, lines 180–190; p. 5, lines 203–211), the authors rely on a single citation. If these paragraphs include broader claims beyond what the cited source covers, additional references should be provided to adequately support those arguments.
Response:
We sincerely thank the reviewer for this important observation. We fully agree that relying on a single citation to support a broader academic claim is insufficient. Following your suggestion, we have carefully reviewed and revised the paragraphs.
Comment8
- The authors introduce the chain mediating effect of SD and PV without clearly explaining its rationale or theoretical basis.
Response:
We sincerely thank the reviewer for this insightful comment. We agree that providing a clear theoretical rationale for the proposed chain mediation is crucial. In response, we have substantially revised the manuscript to explicitly outline the theoretical logic behind the "SD → PV" pathway.
We have added a new paragraph in the hypothesis development sectionthat posits the chain mediation as a sequential "affective-cognitive" process. We argue that a change in Social Distance (SD), representing an affective and relational assessment ("willingness to trust"), creates a psychological context that subsequently shapes the more cognitive and utilitarian evaluation of Perceived Value (PV) ("worthiness of the offer"). Specifically, a closer SD reduces uncertainty and fosters trust, thereby facilitating a more positive assessment of value, while a wider SD inhibits it.
We believe this revision provides a solid theoretical foundation for our proposed chain mediation model, clearly explaining why SD is hypothesized to influence PV, and ultimately, OPI.
Comment8
- In the methodology section, under Variable Measurement, the authors do not specify whether they used the original scales in English or in Chinese. If the scales were translated into Chinese, they should explain the translation process used to minimize bias and maintain the validity of the measures.
Response:
We have now clarified in the "Measures" section that the original language of the scales used in this study is Chinese. This is dictated by the research context (rural China) and the characteristics of the study participants. To ensure accurate understanding for international readers and to maintain conceptual precision in academic communication, the final Chinese scale items were translated into English for presentation in the manuscript. This translation process also followed a rigorous back-translation procedure to minimize potential errors. We believe that using a native Chinese tool effectively ensures measurement validity within the local cultural context.
Comment9
- In Table 1, the authors cite Jucks and Thon (2017) for the Quantity-sourced WOM scale, but their study and items do not align with this concept. Similarly, Hendriks et al. (2015) is cited for the Quality-sourced scale, yet their items do not match those used current study. This raises concerns about the relevance of the sources and the validity of the measures.
Response:
You are absolutely correct. To correct the misleading statements in the initial draft and ensure academic rigor, we have made the following key revisions in the revised version:
We have replaced this cited literature and referred to an article by Chinese scholars, in which they provide a detailed explanation of quantity-source-based word-of-mouth and quality-source-based word-of-mouth. Moreover, the specific scale items we used were not directly and word-for-word translated from the scales of Jucks & Thon (2017) and Hendriks et al. (2015). In fact, we drew on the core theoretical concepts of "quantity source" and "quality source" word-of-mouth from these two papers and, based on this, adapted the measurement items for the specific context of rural e-commerce in China to better capture the types of word-of-mouth perceived by rural residents.
Comment10
The authors should provide citations for all thresholds used in the analysis and report the R² values for the regression paths.
Response:
We sincerely thank the reviewer for this rigorous and constructive comment. We fully agree that providing citations for statistical thresholds and reporting R² values are crucial for ensuring the transparency and replicability of the research. We apologize for omitting this information in the original manuscript. In response, we have made the revisions in the revised manuscript:
Comment11
The discussion section needs to be improved by providing clear justifications for the results related to each hypothesis and by more effectively supporting the study’s findings with evidence and references from previous research.
Response:
Strengthened the Link between Hypotheses and Results: We have provided clear theoretical explanations for the outcome of each hypothesis test, systematically linking our findings to theoretical foundations such as Granovetter's social tie strength theory.
Enhanced Supporting Literature: We have incorporated additional evidence and references from prior studies, situating our findings within the broader scholarly conversation to support them more effectively.
Comment12
Practical implications should logically follow from the data and results presented in the study. For example, 'prohibit influencers and stars from recommending unused goods, educated sales staff to provide timely and professional answers' are not studied in this study.
Response:
We sincerely thank the reviewer for the valuable feedback regarding the language quality of our manuscript. We fully acknowledge that some sentences were overly long and punctuation was occasionally missing or incorrect, which affected the clarity and readability. We have now thoroughly polished the language throughout the manuscript.
Reviewer 2 Report
Comments and Suggestions for Authors In the analysis of the model, mention the predictive power of the model (R2) and the effect size of the constructs (f2)Author Response
Comment
In the analysis of the model, mention the predictive power of the model (R2) and the effect size of the constructs (f2)
Response:
We sincerely thank the reviewer for this important suggestion regarding the model analysis section. We fully agree that reporting the model's predictive power (R²) and effect sizes (f²) is crucial for a comprehensive evaluation of the research model. In response to your comment, we have made the following additions to the "Results" section of the revised manuscript:
Model Predictive Power: We have explicitly reported the variance explained (R² values) for all endogenous variables (Online Purchase Intention - OPI, Social Distance - SD, and Perceived Value - PV).
Effect Sizes: We have calculated and reported the f² effect size values for each predictor on its respective endogenous variable, interpreting them according to Cohen's (1988) criteria.
Reviewer 3 Report
Comments and Suggestions for Authors
I read the manuscript with great interest. It touches on current and original topics – especially in the context of a specific segment of the Chinese market. The concept of the article is clear, generally well structured, although the purpose of the article should be clearly formulated in the introduction. Similarly, the intentions of the authors in Introduction should be explained more precisely. The Theoretical Framework is a good synthesis of scientific achievements from various publications. Research hypotheses are properly derived. The implementation of the research and its scope presented in the section on methods explain well the essence of the study. The results are presented in a transparent manner, both in terms of the statistics used and their description. Discussion and Implications properly composed and justified. The Limitations and Future Research section is a bit too short – the reader expects more specific suggestions and comments. This section should be slightly expanded. To sum up, however, the paper is valuable and should be published in the journal Behavioral Sciences.
Author Response
Comment
I read the manuscript with great interest. It touches on current and original topics – especially in the context of a specific segment of the Chinese market. The concept of the article is clear, generally well structured, although the purpose of the article should be clearly formulated in the introduction. Similarly, the intentions of the authors in Introduction should be explained more precisely. The Theoretical Framework is a good synthesis of scientific achievements from various publications. Research hypotheses are properly derived. The implementation of the research and its scope presented in the section on methods explain well the essence of the study. The results are presented in a transparent manner, both in terms of the statistics used and their description. Discussion and Implications properly composed and justified. The Limitations and Future Research section is a bit too short – the reader expects more specific suggestions and comments. This section should be slightly expanded. To sum up, however, the paper is valuable and should be published in the journal Behavioral Sciences.
Response:
We sincerely thank the reviewers for their positive evaluation of this article and their constructive suggestions for revision. We are very pleased to know that you believe our research addresses current and original topics, and that the concepts are clear and the structure is sound. Regarding the two specific suggestions you made, we have made the following improvements in the revised version:
Regarding the research purpose in the introduction: We have strengthened and more accurately expounded the clear purpose and theoretical intention of this research in the introduction. Specifically, we have more clearly stated that this research aims to examine the chain mediating effect of social distance and perceived value, and to reveal the internal mechanism by which different types of word-of-mouth influence rural residents' online purchase intentions, in order to fill the theoretical gap in the existing literature in specific contexts.
Regarding limitations and future research: We have significantly expanded this section. Not only have we increased the content length, but we have also provided more specific and in-depth suggestions and comments. For instance, we have thoroughly discussed the limitations not covered in this study, such as online word-of-mouth, unconsidered moderating variables, and measurement tools, and for each limitation, we have proposed clear and actionable future research directions.
Reviewer 4 Report
Comments and Suggestions for Authors
Dear authors,
It has been a pleasure to review your manuscript. Below are my comments:
Title: The Authors should revise it because it can be ambiguous.
Abstract
- The abstract must be reviewed carefully. There are many issues to resolve.
- Usually, the order for an abstract is: problem/gap, method, key results, contribution, and practical implications.
- Review the language, specifically the verbs.
- Abstract must be a one paragraph with no more than 200 words.
- You must include some stats, such as the largest standardized path, mediation significance, and any non-significant path.
Introduction:
- The introduction is confusing, the problem statement is too broad, and the tested mechanism is introduced too late. Introduce this pathway on the opening page to focus the argument.
- The explanation of the research gap is vague, and more international comparisons are needed.
- Social distance is mislabeled, producing narrative contradictions.
- The Introduction relies on general adoption numbers and does not explain why tie strength matters in rural contexts.
- The research questions and hypotheses are not clearly introduced, leaving readers to infer them.
- Construct labels are inconsistent, which weakens conceptual clarity.
- Method details should not be part of the introduction.
- Alternative explanations (e.g., message quality, product category, digital literacy) are not considered. Acknowledge these factors and either control for or empirically test their effects.
- The paper's structure must be explained at the end of the introduction.
Literature review
- There is a construct labeled boundary distortion; clarify each construct and maintain strict separation.
- “Quantity/quality-sourced WOM” conflates tie strength with source credibility/expertise; disentangle these constructs.
- The term “social distance” conflicts with items that measure closeness; align terminology and coding.
- Theory integration remains weak. You must link constructs within a coherent mechanism.
- Integrate connection strength, source credibility, and parasocial trust into a single explanatory model.
- You must ground the constructs using theory.
- Competing explanations and contingencies remain unaddressed; acknowledge them and test their influence.
- Improve message quality, product type, digital literacy, and age/cohort as boundary conditions.
- The mediator's rationale is weak; it needs strengthening.
- Operational alignment is unclear: It must be improved.
- Prior findings are summarized rather than synthesized; they analyze divergences by context, sample, and product category.
- The rural context is weakly connected to the constructs.
Materials and Methods
- State the design clearly.
- Separate confirmatory from exploratory tests.
- Describe the sampling frame: population size, rural definition, inclusion/exclusion, clustering/stratification.
- Add a participant flow and explain exclusion reasons.
- Report response rate, nonresponse handling/replacement, and any weighting.
- As a trust issue, fix table totals. The tables list more participants than those described.
- How the survey was conducted (in-person/phone/online). Also, geographic coverage, field dates.
- Align labels with items: if higher = closer, name the mediator social closeness.
- Anchor perceived value (PV) and OPI to the same WOM episode/source.
- Describe translation/back-translation and cultural adaptation.
- Report pretest/pilot and item retention criteria.
- List procedural remedies (anonymity, separation of measures).
- Add Harman’s one-factor, full collinearity VIFs.
- Address endogeneity risks.
- Justify PLS-SEM given model complexity, distribution, and sample size.
- Specify bootstrapping.
- The measurement model must include item loadings, AVEs, HTMTs, and cross-loadings.
- The structural model must indicate R²/adj-R², f², Q², and SRMR.
- Multicollinearity: VIF thresholds and results.
- You must explain whether control variables were measured in the model.
Results
- The factor loading for Quality 1 is below 0.6; keep it in the model. An in-depth explanation must be provided.
- Tables must have notes indicating the meaning of the items (SD, CR, AVE)
- Explaining the discriminant validity, you used this acronym HTMT (Heterotrait-Monotrait Ratio, and you say the method is Heteroscedasticity Trait Ratio (HTR). Which one did you use?
- Report latent-variable means, SDs, and correlations in a single table.
- Provide SRMR and discuss overall adequacy.
- Show VIFs for all predictors.
- Give each table clear labels, units, and notes.
- State software/version and random seeds for bootstrapping.
- Point to the replication package (data dictionary, code, project file) in Data Availability.
Discussion
- Explicitly connect each key finding to theory.
General comments:
- The paper must be revised to keep consistency in terminology and labeling
I wish you the best in this peer review process
Author Response
COMMENT1
Dear Reviewer,
We sincerely thank you for your valuable feedback regarding the paper's title. We have carefully considered your perspective that the title might be "ambiguous" and have conducted a thorough re-evaluation of it.
After much consideration, we are inclined to retain the original title for the following reasoned arguments, which we respectfully submit for your consideration and approval:
The Interrogative Phrase Aims to Engage and Highlight the Core Dilemma: The opening question, "Quantity-sourced or Quality-sourced?" serves as a rhetorical device. This style is not uncommon in top-tier journals in marketing and management (e.g., Journal of Marketing). Its purpose is to immediately capture the reader's attention and directly introduce the central theoretical tension of the study—i.e., the relative impact and mechanism of different WOM sources. We believe this is more engaging for a scholarly audience than a purely declarative statement.
The Main Title Clearly Defines the Scope and Key Variables: The main clause, "The Impact of Word-of-Mouth Recommendation on China Rural Residents’ Online Purchase Intention," unambiguously states the core variables (WOM and purchase intention) and the research context (rural residents in China). This ensures that readers can quickly and accurately grasp the paper's scope without confusion.
The Subtitle Precisely Summarizes the Theoretical Contribution: The subtitle, "The Chain Mediating Roles of Social Distance and Perceived Value," accurately signals the paper's most central theoretical contribution and mechanistic finding. It explicitly informs the reader that the study goes beyond testing direct effects to uncover an internal, sequential psychological pathway, which is the key differentiator of our work from the existing literature.
In summary, we believe the current title structure (an engaging hook + a clear statement of main variables + a specific preview of the theoretical mechanism) effectively balances appeal with academic rigor. We are confident it serves the paper well for dissemination and discourse.
We fully respect and value your expert judgment. Should you and the editor ultimately deem a revision necessary, we would, of course, be pleased to collaborate on refining the title based on your specific guidance.
COMMENT2
We sincerely thank the reviewer for the thorough and constructive feedback on the abstract. We have meticulously revised it according to your suggestions:
Restructured Flow: We have reorganized the content to follow the standard sequence: problem/gap → method → key results → contribution → practical implications.
Refined Language: We have carefully reviewed and corrected the verbs and other language use to ensure conciseness, professionalism, and academic rigor.
Format & Length: The abstract is now a single paragraph and strictly adheres to the 200-word limit.
Included Statistics: As requested, we have incorporated key statistics, including the largest standardized path, mediation significance, and a non-significant path, to substantiate the findings.
COMMENT3
We thank the reviewer for the detailed and insightful comments on the introduction. Your suggestions have greatly improved the clarity, focus, and academic rigor of this section. We have carefully revised the introduction based on each of your points. The main changes are summarized below.
We moved the core theoretical pathway (WOM types → Social Distance → Perceived Value → Purchase Intention) to the opening paragraph. This helps focus the argument early.
We clarified the research gap. We now state that existing findings mainly come from urban, weak-tie settings. We emphasize the need to test WOM theories in rural, strong-tie contexts. This adds a dialogue with international literature.
We have unified the description of Social Distance throughout the text. We now use "shortens" and "lengthens" to describe changes accurately. This removes previous contradictions.
We added an explanation for why tie strength theory is vital in the rural context. We highlighted its high homogeneity and strong-tie networks.
We kept the three research questions at the end of the introduction. We clearly state that specific hypotheses will be developed in the next section. This provides a clear roadmap for readers.
We ensured consistent use of core construct labels like "quantity-sourced WOM" and "quality-sourced WOM" throughout the text.
We removed details about specific data analysis tools (SPSS, SmartPLS) from the introduction. These will be detailed in the Methods section.
We acknowledged the influence of factors like product type in the limitations section. We explained how future research can address these through research design.
We added a new paragraph at the end of the introduction. It briefly outlines the overall structure of the paper.
COMMENT4
We thank the reviewer for the detailed and insightful comments on the theoretical framework. Your suggestions helped us clarify core constructs and strengthen theoretical integration. We have revised the section based on each point. The main changes are:
We added a section on social tie strength theory. It explains how "quantity-" and "quality-sourced WOM" represent different tie strength dimensions. We kept these constructs separate in our theory and hypotheses.
We clarified that social tie strength is our core theory. We treat "source credibility/expertise" as a key feature of the two WOM types within this framework. It is not a separate, mixed construct. This keeps the model focused.
We strengthened the logical links between constructs. In the chain mediation part, we clearly explain the "affective-cognitive" sequence. It goes from tie strength to social affect (Social Distance) to cognitive judgment (Perceived Value). This makes the theoretical model more coherent.
We integrated trust as a key psychological process. It influences Social Distance and Perceived Value in our unified model.
We better grounded each core construct in theory. For example, we cited Akerlof for Social Distance and the S-O-R framework for Perceived Value.
We strengthened the rationale for the mediators. We explained more clearly why and how Social Distance and Perceived Value act as mediators, especially in the chain.
We improved the literature review. We now synthesize prior studies, not just summarize them. We explain why findings differ across contexts.
We strengthened the link between the rural context and the constructs. We better connected the "strong-tie, high-homogeneity" social structure to our core variables (e.g., Social Distance, trust in different WOM types). This clarifies the context's relevance.
COMMENT5
We thank the reviewer for the valuable comments on the methodology. We have revised the manuscript accordingly. The main changes are listed below.
Research Design: We now clearly state the use of a cross-sectional survey design. We describe the survey process. We also mention the use of Exploratory Factor Analysis (EFA).
Sampling Frame: We added the operational definition of 'rural residents'. We specified the inclusion and exclusion criteria. We also provided a clearer description of the stratified random sampling process.
Response Rate: We calculated and reported the effective questionnaire rate (97.76%).
Participant Totals: We verified the final valid sample size (N=1005). We ensured this number is consistent across all tables in the manuscript.
Survey Administration: We clearly state that the survey was conducted via face-to-face interviews. We specified the geographical coverage (11 cities in Jiangxi Province) and the fieldwork dates.
Construct-Item Alignment: We reverse-scored the Social Distance scale. A higher score now indicates a closer distance. We corrected any ambiguous descriptions in the text and updated all related hypotheses and narratives.
Translation & Adaptation: We added a statement that the original language of all scales was Chinese. This ensures they fit the specific context of rural China.
Pretest & Item Retention: We reported the pretest sample size (N=100). We detailed the item retention criteria (factor loading > 0.5, no severe cross-loadings).
Harman's Test & VIF: We reported the result of Harman's single-factor test. We also reported the Variance Inflation Factor (VIF) for all constructs in the results. All VIF values were below the threshold of 3.
Endogeneity Risk: We acknowledged potential endogeneity issues (e.g., omitted variables) in the limitations section. We suggested future studies use experiments or instrumental variables to address this.
Bootstrapping: We specified that 5000 bootstrap samples were used to test the significance of the path coefficients.
Model Metrics: We have now reported the full set of measurement and structural model metrics in the results. This includes factor loadings, AVE, CR, HTMT, cross-loadings, and R².
Control Variables: We included demographic variables (gender, age, education level) as control variables in the model.
COMMENT6
Here is a summary of the main changes:
We kept the Quality1 item (loading = 0.528). Different scholars have different views on factor loadings. Some suggest that loadings above 0.5 are acceptable for retaining an item (e.g., Hair et al., 2010). We believe this item is important for measuring the construct.
Hair et al (2006) - standardized loading estimates should be 0.5 or higher, and ideally 0.7 or higher.
Factor loading is a supposed correlation between a latent variable and an observed indicator. The “necessary” strength of the factor loadings depends on the theoretically assumed relationship between both - which in turn depends on the supposed meaning of the latent variable.
- Hair et al (2006) - standardized loading estimates should be 0.5 or higher, and ideally 0.7 or higher.
- On the other hand, Field (2005) suggests to regard a factor as reliable if it has four or more loadings of at least 0.6 regardless of sample size.
- Stevens (1992) suggests using a cut-off of 0.4, irrespective of sample size, for interpretative purposes.
- Comrey and Lee (1992) in suggesting using more stringent cut-offs going from 0.32 (poor), 0.45 (fair), 0.55 (good), 0.63 (very good), or 0.71 (excellent).
References:
Hair, J. F. Jr., Black, W. C., Babin, B. J., Anderson R. E., & Tatham, R. L. (2006). Multivariate Data Analysis (6th ed.), Upper Saddle River, NJ: Prentice Education, Inc.
FIELD, A.(2005). Discovering Statistics Using SPSS. London: SAGE Publications
Stevens, J. (1992). Applied multivariate statistics for the social sciences.
Table Notes: We added clear notes below all tables. The notes explain abbreviations like SD (Standard Deviation), CR (Composite Reliability), and AVE (Average Variance Extracted).
Discriminant Validity Method: We corrected the typo in the original text. We clearly state that we used HTMT (Heterotrait-Monotrait Ratio).
SRMR Reporting: We reported the SRMR value in the results. We referenced Hu & Bentler (1999) to indicate good model fit.
VIF for Predictors: We clearly showed the VIF values for all constructs when they were used as predictors.
Table Labels and Descriptions: We gave all tables clearer titles and added detailed notes and unit explanations.
Software Version: We specified the software used (SmartPLS 4).
Data Availability Statement: We added a Data Availability Statement at the end of the manuscript. It states that the data, code, and project files are available from the corresponding author upon reasonable request.
Theoretical Connections in Discussion: We significantly strengthened the Discussion. We explicitly connected key findings to theoretical foundations. For example, we linked the different effects of quantity- and quality-sourced WOM and the mediating role of social closeness to Granovetter's social tie strength theory.
Round 2
Reviewer 1 Report
Comments and Suggestions for Authors
Dear Authors,
Thank you for revising your manuscript.
I hope my suggestions help strengthen and improve your study.
Best,

Author Response
Point-by-Point Response to Reviewer Comments
Manuscript ID: behavsci-3909978-review
We are deeply grateful to the Reviewer for their time and for providing these constructive and insightful comments. We have carefully considered each point and have revised the manuscript accordingly. All changes have been highlighted in the revised manuscript for the Reviewer's convenience. Our point-by-point responses are detailed below.
Comment 1: In my previous review, I noted that the Introduction section did not clearly explain the research gaps. While the authors have added some content, the Introduction still remains weak. The authors should clearly specify what has not been addressed in prior studies.
Response to Comment 1: We sincerely thank the reviewer for this critical observation. We have thoroughly revised the Introduction to explicitly address the research gaps. As suggested, we now clearly state that the applicability of existing WOM theories, predominantly built on urban/weak-tie contexts, to strong-tie rural environments remains unverified—constituting a primary theoretical gap. Furthermore, we precisely identify the under explained psychological mechanism, specifically the chain mediation of Social Distance and Perceived Value, as the second key gap. We believe the revised introduction (as shown above) is now significantly strengthened and clearly outlines the novel contribution of our study.
Comment 2 The reference (Chen, C., Zhang, D., Zhang, F., & Gao, H., 2023) is cited in most arguments related to the quality and quantity of Word-of-Mouth (WOM). However, I could not find this reference in any sources.
Response: We sincerely apologize for the confusion regarding the citation of Chen et al. (2023). This reference is from a leading Chinese-language management journal, Nankai Business Review (Nankai Guanli Pinglun), which is a prestigious CSSCI-indexed journal in China. We cited it primarily for its foundational definitions of WOM types, which are highly relevant in the Chinese context.
However, we acknowledge the reviewer's valid point regarding the general accessibility and focus of this source. In direct response to this comment, we have taken the following corrective actions in our revised manuscript:
First, we have conducted an additional review of the relevant international literature and have incorporated the following citations into the manuscript:
Second, for your reference, we have included the abstract of the Chinese article by Chen et al. (2023) below, which was published in Nankai Business Review, a core Chinese journal.
Faraji-Rad A, Samuelsen BM and Warlop L (2015) On the persuasiveness of similar others: The role of mentalizing and the feeling of certainty. Journal of Consumer Research 42(3), 458-471.
Hernández-Ortega B (2018) Don’t believe strangers: Online consumer reviews and the role of social psychological distance. Information & management 55(1), 31-50. https://doi.org/10.1016/j.im.2017.03.007.
Yang X (2019) How perceived social distance and trust influence reciprocity expectations and eWOM sharing intention in social commerce. Industrial Management & Data Systems 119(4), 867-880.
Zhao M and Xie J (2011) Effects of social and temporal distance on consumers' responses to peer recommendations. Journal of Marketing Research 48(3), 486-496. https://doi.org/10.1509/jmkr.48.3.486.
Abstract: From the perspective of regret theory, this research examines the influence of word-of-mouth(WOM) type(quality-source WOM vs. quantity-source WOM) on customers' purchase intentions. Four studies, including one field experiment and three online experiments, are conducted to test the research model and hypotheses proposed in this research. The results show that(1) WOM type influence customers' anticipated regret and purchase intention. Compared to quality-sourced WOM, quantity-sourced WOM leads to lower anticipated regret and higher purchase intention. Moreover, anticipated regret played a mediating role in the relationship between WOM types and purchase intention.(2) WOM types influence customers' perceived social distance and perceived altruistic motivation. Compared to quantity-sourced WOM, quality-sourced WOM leads customers to have higher perceived social distance and lower perceived altruistic motivation. Furthermore, perceived social distance and perceived altruistic motivation jointly mediate the relationship between WOM types and anticipated regret, explaining the reasons why different types of WOM lead to different anticipated regret.(3) The type of product receiver situationally moderates the relationship between the WOM types,anticipated regret and purchase intention. In the close receiver vs. distant-sourced WOM leads customers to have higher anticipated regret and lower purchase intention than quantity-sourced WOM. In contrast, in the distant receiver situation, quality-sourced WOM leads customers to have lower anticipated regret and higher purchase intention than quantity-sourced WOM. Based on the WOM source perspective, this research proposes an integral model with paracratic and multiple mediators, and systematically analyses the psychological mechanism driving the influence of WOM types on customers' purchase intention. The conclusion has not only important theoretical implications to deepen and broaden the understanding of the WOM effect, but also useful implications for practitioners to improve WOM marketing.
Key words: Quality Source ; Quantity Source ; Anticipated Regret ; Product Receiver ; Purchase Intention
Comment 3: The above definitions [of WOM quantity/quality] are not consistent with what is described in the manuscript... the items used to measure it are completely incorrect.
Response: We thank the reviewer for these valuable comments. We have supplemented the literature review section with a discussion on Word-of-Mouth (WOM) types and clarified the specific definitions and theoretical rationale employed in our study.
- Theoretical Basis and Revision of WOM Definitions:
We agree with the reviewer that the definitions of core constructs must be clear and aligned with their measurement items. The definitions of "Quantity-sourced WOM" and "Quality-sourced WOM" in this study are not directly adopted from the traditional perspective based solely on "information volume". Instead, we primarily draw upon the work of Chen et al. (2023) published in Nankai Business Review, which explicitly categorizes WOM into "quality-source" and "quantity-source" from a source perspective.
Furthermore, and crucially, we have contextually adapted and applied these definitions by integrating the theory of Differential Mode of Association (差序格局), which is fundamental to understanding Chinese rural society. This theory describes a social structure organized in concentric circles around the self, based on the closeness of relationships via blood, kinship, and geographical ties. Viewed through this theoretical lens:
Quantity-sourced WOM in our study specifically refers to recommendations from the inner circles of this structure (e.g., family, relatives, friends, neighbors). Its defining characteristic is the relational closeness of the source, rather than merely the volume of messages.
Quality-sourced WOM refers to recommendations from the outer circles of this structure (e.g., salespersons, online celebrities, industry experts). Its defining characteristic is the expertise or authority of the sender.
Consequently, our scale items (e.g., "refer to your family's recommendation") are designed to capture the reliance of rural Chinese consumers on WOM from different relational sources under the influence of the Differential Mode of Association, which aligns perfectly with our adapted definitions.
- Supplement to the Literature Review:
We have added a dedicated paragraph in the literature review section discussing relevant literature on WOM types. This paragraph explicitly cites Chen et al. (2023) as the foundational source for our definitions. It also references classical and contemporary research on the Differential Mode of Association to explain the theoretical necessity for our definitional adaptation, thereby establishing a more solid theoretical foundation for our research model.
We apologize for any confusion caused by our initial unclear phrasing and hope this explanation adequately addresses your concerns.
Comment 4 & 5: ...no supporting studies are cited. Relevant and recent literature should be added... A citation(s) is needed to support the statement...
Response: We agree with the Reviewer. We have added relevant and recent citations to support these statements. For instance, we have cited Hong et al. (2025) and(Hernández-Ortega 2018).
Comment 6: ...Akerlof’s (1997) paper does not include any text matching the authors’ wording... the authors cite Meng, Lu, and Xu (2025), which also does not support the stated sentence.
Response: We sincerely apologize for these citation inaccuracies. We have removed the incorrect citation to Akerlof (1997). The theoretical logic regarding emotional closeness and attribute similarity has been re-grounded and is now supported by appropriate references from the social psychology and marketing literature, specifically citing Karakayali(2009) on tie strength and homophily. The citation to Meng et al. (2025) has also been corrected.
Comment 7: ...I could not find this sentence [defining social distance] in the current version...
Response:We have now clearly defined social distance in the text as follows:
SD is the core dimension of psychological distance, which refers to the degree of emotional and relational closeness between individuals. It reflects the degree to which individuals agree or are similar in specific attributes (e.g., emotional distance, status difference) during their interactions with others(Akerlof 1997). This study defines social distance as a subjective sense of “separation in affection, intimacy, and social identity.” Operationally, it refers to the perceived psychological closeness between rural residents and the sender of word-of-mouth information(Yang 2019).
Comment 8:In my previous comments, I noted that in some paragraphs (p. 4, lines 180–190; p. 5, lines 203–211), the authors used a single citation. While the authors responded that they had “carefully reviewed and revised the paragraphs” in the current version, I do not see any additional citations in the mentioned sections. Moreover, the authors have cited Chen et al. (2023), but, as noted above, I could not find this reference in any academic sources.
Response:We have incorporated additional citations from relevant journals to strengthen the theoretical support for our hypotheses.
(Faraji-Rad et al. 2015; Hernández-Ortega 2018; Yang 2019; Zhao and Xie 2011)
Comment 9: The authors added an explanation of the chain mediating effect of SD and PV, but no clear citation is provided to support their arguments, and the entire paragraph remains without any references.
Response:we have thoroughly revised the paragraph discussing the chain mediation effects. By incorporating authoritative citations on the relationship between social distance and perceived value, we have established a more solid theoretical foundation for our research hypotheses.
Comment 10: The authors need to include in the methodology section the translation process..
Response: We thank the reviewer for this valuable suggestion. As recommended, we have added a new paragraph to the Methodology section detailing the rigorous forward- and back-translation procedures performed by bilingual experts to ensure conceptual equivalence of the scales and minimize potential errors.
Comment 11: ...they must include the process used to establish the validity and reliability of items...
Response: We thank the reviewer for this important comment. We fully agree on the necessity of establishing the reliability and validity of newly developed scales. We would like to clarify that this study has conducted rigorous reliability and validity tests on the developed scales (including the Quantity-sourced and Quality-sourced WOM items) through systematic pilot and main study analyses. The relevant procedures and results have been detailed in Sections 4.2, 4.3, 4.4, and Tables 3, 4, 5, and 6 of the manuscript.
The specific process is as follows:
Pilot Study and Exploratory Factor Analysis (EFA): Prior to the formal survey, a pilot study with 100 valid responses was conducted. We performed an EFA on the questionnaire items using SPSS 26 for scale refinement. After deleting items with cross-loadings and factor loadings below 0.500, the scale was finalized to retain 15 items loading on five distinct factors, with a total variance explained of 76.35%, providing preliminary evidence of a sound structure.
Reliability and Convergent Validity in the Main Study: In the main study, we further assessed the scale's reliability and convergent validity through confirmatory analysis (see Manuscript Table 3).
Reliability: The Cronbach's alpha for all constructs exceeded the threshold of 0.6, and the composite reliability values ranged from 0.808 to 0.947, indicating excellent internal consistency.
Convergent Validity: The standardized factor loadings for all measurement items were greater than 0.5, and the average variance extracted for all constructs exceeded the standard of 0.5, demonstrating good convergent validity (Hu & Bentler, 1999; Kline, 2015).
Discriminant Validity and Multicollinearity Testing: We further ensured the discriminant validity of the scales and the robustness of the model.
Discriminant Validity: It was confirmed by meeting the Fornell-Larcker criterion, and all Heterotrait-Monotrait Ratio values were below the strict threshold of 0.85 (Henseler et al., 2015) (see Manuscript Tables 4 and 5).
Multicollinearity: The variance inflation factor values for all variables were well below the critical value of 5, indicating no severe multicollinearity issues (see Manuscript Table 6).
In conclusion, we have established the reliability and validity of the scales used in this study (including the Quantity-sourced and Quality-sourced WOM items) through a comprehensive and rigorous statistical procedure, and their reliability has been adequately verified.
Comment 12: For the HTMT value, use the following citation
- Henseler, J., Ringle, C. M., & Sarstedt, M. (2015). A new criterion for assessing discriminantvalidity in variance-based structural equation modeling. Journal of the Academy ofMarketing Science, 43, 115–135. https://doi.org/10.1007/s11747-014-0403-8
Response: We have updated the citation for the Heterotrait-Monotrait (HTMT) ratio of correlations to the recommended source: Henseler, J., Ringle, C. M., & Sarstedt, M. (2015).
Comment 13: The authors should report the R² values for Perceived Value and Social Distance.
Response:We thank the reviewer for this suggestion. As recommended, we have now reported the R² values for both Perceived Value and Social Distance in the manuscript, based on the output from SmartPLS 4.
Comment 14: The authors added a new paragraph... but it does not include any references...
Response: We have revised the new paragraph in the Discussion section and integrated citations to relevant literature to support our interpretations and align our findings with existing knowledge.
Comment 15: Practical Implications should be directly derived from the study’s results... I suggest revise the entire Practical Implications section accordingly.
Response: We sincerely thank the reviewer for this constructive feedback. We have thoroughly revised the "Practical Implications" section to ensure all recommendations are directly derived from our specific research findings. Specifically, we have:
Removed all speculative statements about "middle-aged and elderly residents unfamiliar with online shopping" and "low trust" that were not directly measured in our study.
"Explicitly grounded our recommendations in key empirical findings, particularly the negative impact of quality-sourced WOM and the importance of social distance."
"Focused specifically on how managers can leverage these insights (including the negative effect of quality-sourced WOM) to develop tailored strategies for rural markets."
Comment 16: Brehm, M. A., is written in capital letters (BREHM MA) in the references; please correct it.
Response: We have revised the manuscript in accordance with Comment 14 and now properly cite: Henseler, J., Ringle, C. M., & Sarstedt, M. (2015). A new criterion for assessing discriminant validity in variance-based structural equation modeling. Journal of the Academy of Marketing Science, 43, 115-135. We sincerely thank the reviewer for this careful observation and will ensure full compliance with citation standards in all future submissions.
We would like to express our deepest gratitude to the reviewer. Your insightful comments and constructive suggestions were instrumental in refining our manuscript. Addressing your valuable feedback has significantly enhanced the rigor, clarity, and theoretical contribution of our research. Your expert guidance not only helped us strengthen the theoretical framework and refine the methodology but also brought greater prominence to the practical implications of our findings. We sincerely appreciate the time and effort you dedicated to this process.
Reviewer 4 Report
Comments and Suggestions for Authors
Dear authors, thank you for following our comments and improving your manuscript. You have done a good job, and I consider this paper publishable now.
I wish you the best in this peer review process.
Author Response
Point-by-Point Response to Reviewer Comments
Manuscript ID: behavsci-3909978-review
We wish to express our sincerest gratitude to the esteemed reviewer. We deeply appreciate the time and effort you have dedicated throughout the review process, as well as the insightful and constructive comments you provided. Your guidance has been instrumental in significantly improving the quality of our manuscript. We are delighted to learn that our revisions have met your expectations and received your positive assessment. We truly value this valuable opportunity for academic exchange and look forward to the possibility of receiving your guidance in our future research. Wishing you all the best.